# Polymorphisms and drug resistance analysis of HIV-1 isolates from patients on first line antiretroviral therapy (ART) in South-eastern Nigeria

**Augustine O. Udeze**[1,2], **David O. Olaleye**[1], **Georgina N. Odaibo**[1] *

**1** Department of Virology, College of Medicine, University of Ibadan, Ibadan, Nigeria, **2** Virology Unit, Department of Microbiology, University of Ilorin, Ilorin, Nigeria

* foreodaibo@hotmail.com

**Data Availability Statement:** Nucleotide sequences of the isolates obtained in this study were deposited in GenBank under assigned accession numbers MF458138- MF458165.

## Abstract

Acquisition of resistance mutations by HIV-1 isolates causes treatment failure among infected patients receiving antiretroviral therapy (ART). This study determined patterns of drug-resistance mutations (DRMs) among HIV-1 isolates from patients receiving first-line ART in South-eastern Nigeria. Blood samples were collected from HIV-1 infected patients accessing antiretroviral treatment centers at General Hospital Awo-Omamma, Imo state, State Hospital Asaba, Delta state and St Joseph's Catholic Hospital Adazi, Anambra state and used for HIV-1 DNA sequencing and phylogenetic analysis. DRMs were scored using combination of Stanford algorithm and the 2015 International Antiviral Society-USA list while drug susceptibility was predicted using Stanford algorithm. Twenty eight of the HIV-1 isolates were sequenced and identified as subtypes G (35.7%), CRF02_AG (57.1%) and unclassifiable, $U^G$ (7.1%). Major PI resistance-associated mutations were identified at two sites including M46L (16.7% of subtype G/$U^G$) and V82L (6.3% of CRF02_AG). Minor PI resistance-associated mutations identified among subtype G/$U^G$ are L10V/I (8.3%) and K20I (100%) while L10V/I (50%), K20I (100%), L33F (6.3%) and N88D (6.3%) were identified among CRF02_AG. Other polymorphisms found include; I13V/A, E35Q, M36I/L, N37D/S/E/H, R57K/G, L63T/P/S/Q, C67E/S, H69K/R, K70R, V82I and L89M in the range of 28.6% to 100% among the different subtypes. Interpretation based on Stanford algorithm showed that Darunavir/ritonavir is the only regimen whose potency was not compromised by the circulating mutations. Identification of major and minor PI resistance mutations in this study underscores the need for drug resistance testing prior to initiation of second line antiretroviral therapy in Nigeria.

## Introduction

Human immunodeficiency virus type-1 (HIV-1) is characterized by high level of genetic diversity with the distribution of the different variants varying by regions globally [1]. Due to the

**Funding:** The author(s) received no specific funding for this work.

**Competing interests:** The authors have declared that no competing interests exist.

unstable nature of its genome, new variants continue to emerge especially in areas with circulating multiple subtypes [2].

Despite increasing availability of antiretroviral (ARV) drugs, the genetic diversity posed a major challenge to global management of HIV infection. The use of Highly Active Antiretroviral Therapy (HAART) proved highly effective yet treatment failure remains a common occurrence among patients. In addition to adherence issues [3], emergence of drug resistant variants has been identified as a major obstacle to the effectiveness of antiretroviral therapy (ART) and one of the leading causes of treatment failure [4].

Emergence of drug resistance variants of HIV-1 has been attributed to mutations within the HIV-1 *pol* genes that encode the molecular targets for major ARV drugs [5]. A number of factors are believed to contribute to the acquisition of drug resistance in Africa including; lack of plasma viral load monitoring [6], drug interactions [7], treatment interruptions due to drug stock-outs [8] and the use of substandard antiretroviral regimens [9]. Available data shows that effectiveness of ARV therapy is also influenced by both viral subtype and pre-existing mutations [10, 11]. Furthermore, it has been postulated that the pathways to drug resistance may be affected by pre-existing polymorphisms among different HIV-1 subtypes [12].

Most reports on HIV-1 drug resistance so far has focused on subtype B viruses which is prevalent in the Western world. There is however comparatively little available data from less developed countries where non-B subtypes predominate. In Nigeria where the epidemic is largely driven by non-B subtypes, reports on HIV drug resistance and polymorphisms [12–21] have primarily focused on resistance to non-nucleoside reverse transcriptase inhibitors (NNRTIs) and nucleoside reverse transcriptase inhibitors (NRTIs) while resistance to Protease inhibitors (PI) remain understudied. Since the commencement of ART program in Nigeria in 2001, government has collaborated with some donor agencies such as Global Fund to Fight AIDS, Tuberculosis, and Malaria and US President's Emergency Plan for AIDS Relief (PEPFAR) to scale up its ART clinics. With subsequent revision of the treatment guidelines by WHO first in 2010 [22], 2013 [23] and more recently in 2016 [24], initiation of ART for infected individuals is now recommended regardless of WHO clinical stage and at any CD4 cell count as against the previous $\leq$200 cells/mm$^3$ during the pre-2010 era. This greatly increased the number of patients commencing first-line ART with anticipated increase in development of drug resistance. In Nigeria the recommended first line ARV drugs between 2010 and 2013 were AZT+3TC +EFV OR AZT+3TC+NVP OR TDF +3TC (or FTC) + EFV OR TDF +3TC (or FTC) + NVP. Patients failing first-line ARV treatments require switching to second-line regimens. Drug-regimens consist mostly of NNRTIs and NRTIs in the first-line with the addition of protease inhibitors (PIs) in the second-line. Adequate knowledge of drug resistance mutations and polymorphisms in *protease* gene of the circulating strains is therefore needed to help optimize the selection of second-line regimens for patients who are failing first-line regimens and limit the acquisition of cross-resistance. The aim of this study was to characterize and determine the polymorphisms and drug resistance mutations to PIs of HIV-1 isolates from first-line ART-experienced individuals in South-eastern Nigeria.

## Materials and methods

### Study participants and sample collection

The study participants included 28 HIV-1-infected individuals assessing therapy at HIV clinics located in General Hospital Awo-Omamma, Imo state; State Hospital Asaba, Delta state and St Joseph's Catholic Hospital Adazi, Anambra state between February and May 2012. They consisted of 11 males and 17 females with mean age of 34.7 years (range: 25–50 years). HIV infected patients who are receiving treatment are included in the study while drug naïve patients are excluded. About 5ml of venous blood samples were collected from each participant for the study after

informed consent. The study protocol was approved by University of Ibadan/UCH ethical review board (UI/EC/11/0178). Due to high level of patients with no formal education, option of verbal/oral consent was adopted as the ethics committee was not specific on mode.

### DNA extraction, nested PCR, sequencing and phylogenetic analysis

Genomic DNA was extracted from the samples using modified phenol-chloroform extraction procedure and precipitated using ethanol. Nested polymerase chain reaction was used to amplify a 524-bp fragment of the *pol* gene from the extracted DNA. The first round PCR primers were OJ1 (5′–AAATGATGACAGCATGTCAGGGAG–3′; HXB2, 1823–1846) and OJ2 (5′– TATCTACTTGTTCATTTCCTCCAAT–3′; HXB2, 4173–4197) while the second round primers were OJ3 (5′–AGACAGGCTAATTTTTTAGGGA–3′; HXB2, 2074–2095) and OJ4 (5′–CATTCCTGGCTTTAATTTTACTGG–3′; HXB2, 2574–2597) [12]. The PCR products were separated by agarose gel electrophoresis. The amplicons were purified using WIZARD Purification Kit (Promega) according to manufacturer's protocol. The *protease* gene was sequenced using Big Dye Terminator Cycle Sequencing Ready Reaction kit v3.1 (Applied Biosystems, Foster City, CA, USA) with primers OJ3 and OJ4 as sequencing primers. Sequences were generated using ABI Prism 3130 XL genetic analyzer (Applied Biosystems, California, USA).

The sequences were aligned with HIV-1 *protease* reference sequences of various subtypes downloaded from the Los Alamos HIV Sequence Database (www.hiv.lanl.gov). Phylogenetic inferences were performed by the neighbour-joining method with 1,000 bootstrap replicates under Kimura's two-parameter correction using MEGA 6.06. The evolutionary distances were computed using the Maximum Composite Likelihood method and are in the units of the number of base substitutions per site [25]. Sequences have been deposited in the GenBank with accession numbers MF458138- MF458165.

### Drug resistance mutation analysis and prediction of susceptibility

The nucleotide sequences were translated to amino acid sequences using MEGA 6.06 software. The whole *protease* gene was analyzed to identify potential drug resistance mutations (DRMs), polymorphisms at DRM sites, and subtype-specific polymorphisms. DRMs were classified as minor or major base on the September 15, 2016 updated HIV drug resistance data base (http://hivdb.stanford.edu) and the latest definition of the International Antiviral Society (IAS-USA) mutation lists updated in 2015 [26]. Possible impact of the DRMs on the therapeutic response was predicted by use of Stanford drug-resistance algorithm.

## Results

### Phylogenetic analysis of the sequences

Phylogenetic analysis revealed that 10 (35.7%) and 16 (57.1%) of the virus isolates were HIV-1G and CRF02_AG respectively while 2 (7.1%) sequences were unclassifiable. Blast results of sequences of these two isolates from the Los Alamos HIV-1 sequence database also showed that the isolates had closest similarity to HIV-1 subtype G and are hereby referred to as unclassified subtype G (U$^G$) (Fig 1).

### Amino acid diversity of the *protease* region

The amino acid alignment of the samples with subtype B consensus (Cons B) is shown in Fig 2. The sequence analysis of the *Protease* showed total variation in 41 out of the 99 amino acid positions (41.4% of variation) when compared to Cons B. There were no insertions or deletions in the sequence. High variation was observed for amino acid positions I13(100.0%), K14

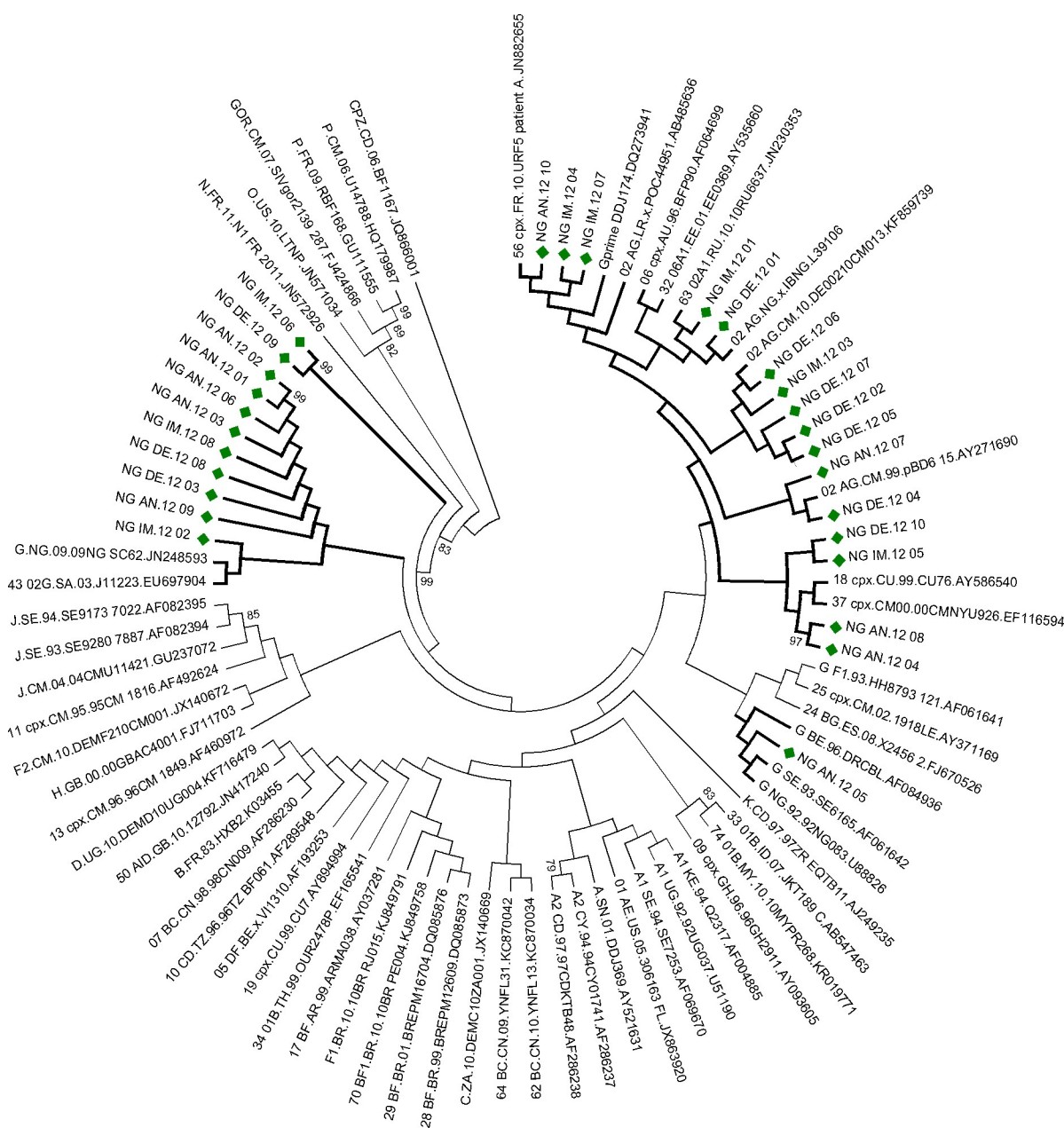

**Fig 1. Phylogenetic tree of study sequences aligned with sequences of reference subtypes from Los Alamos database.** Multiple sequences alignment and phylogenetic tree were constructed using ClustalW and neighbour-joining algorithm with Maximum Composite Likelihood model in MEGA 6.06 software. Statistical significance of the tree topology was tested by 1000 bootstrap replication. Only bootstrap values >70% are displayed at the nodes. Study sequences are marked with solid blocks.

(48.4%), K20(100.0%), E35(60.7%), M36(100.0%), R41(89.3%), R57(42.9%), L63(53.6%), C67 (42.9%), H69(100.0%), V82(46.4%) and L89(100.0%) among the isolates in this study.

## Mutations and polymorphisms at previously characterized drug resistance sites

Major drug resistance mutations were identified at two *protease* sites (M46L and V82L) previously characterized for drug resistance in three of the sequences (Table 1). Polymorphisms at

**Fig 2. Alignment of protease amino acid sequences of the isolates compared with the subtype B consensus (cons B) sequence.** Amino acids are represented by the single-letter amino acid code. Each amino acid residue not differing from the reference sequence is represented by a dot.

known secondary mutation sites (K20I, M36I/L, H69K/R and L89M) were found in all the samples while L63T/P/S/Q was found in 83.3% (10/12) and 31.3% (5/16) of subtypes G/U$^G$ and CRF02_AG respectively. Furthermore, a polymorphism at a known primary mutation site (V82I) was found in all subtype G/U$^G$ samples. Other common mutations at positions not selected for drug resistance include; I13V/A which occurred in all the samples; C67E/S found in all subtypes G/U$^G$; E35Q occurred in 91.7% of subtypes G/U$^G$; N37D/S/E/H which occurred in 41.7% and 25.0% of subtypes G/U$^G$ and CRF02_AG respectively. Also R57K/G occurred in 83.3% (10/12) of subtypes G/U$^G$ and 12.5% (2/16) of CRF02_AG. The frequency of occurrence of the different mutations and/or polymorphisms is shown in Table 2. It is important to note that C67E/S, V82I and E35Q mutations were found only among the G/U$^G$ isolates.

## Drug resistance analysis

Major mutation that confers resistance to protease inhibitors, M46L was found in two out of the twelve (16.7%) subtypes G/ U$^G$ sequences while V82L, was present in one out of the sixteen (6.3%) CRF02_AG sequences. Minor PI mutations detected among the isolates include; L10I/V [7/28 (25.0%)], K20I [28/28 (100%)], L33F [1/28 (3.6%)] and N88D [1/28 (3.6%)]. The different PI-resistance mutations and the patterns of resistance to the different PIs are shown in Table 3.

Fig 3 shows the frequency of occurrence of the predicted viral susceptibility of the isolates to PIs. Of these virus sequences harbouring ≥1 DRM, susceptibility to boosted Darunavir was maintained in all (100%) of the isolates. Reduced susceptibility was predicted for boosted Atazanavir, Indinavir and Lopinavir in about 11% of sequences, Saquinavir in 4%, Fosamprenavir and Tipranavir in 14%. Reduced HIV-1 susceptibility was predicted for the only non-boosted Nelfinavir in all (100%) of the sequences due to general presence of K20I.

## Discussion

The frequency of occurrence (10.7%) of major PI resistance mutations, M46L and V82L, obtained in this study is somewhat lower than the 39.1% recorded in a similar study conducted

**Table 1. Protease mutations/polymorphisms detected among HIV isolates in South-Eastern Nigeria.**

| Sample | Subtype | Mutations/Polymorphisms |
|---|---|---|
| NG_AN.12_01 | G | I13V, K14R, *K20I*, E35Q, M36I, R41K, K43R, **M46L**, R57K, Q61X, L63T, C67E, H69K, K70R, V82I, L89M |
| NG_AN.12_02 | G | *K20I*, I13V, K14R, E35Q, M36I, R41K, K43R, **M46L**, R57K, L63T, C67E, H69K, K70R, V82I, L89M |
| NG_AN.12_03 | G | I13V, *K20I*, E35Q, M36I, R41K, R57K, L63P, C67E, H69K, V82I, L89M |
| NG_AN.12_04 | CRF02_AG | *L10V*, I13V, L19P, *K20I*, E35D, M36I, R41K, R59G, H69K, L89M, G94R, C95W, N98H |
| NG_AN.12_05 | G | I13V, *K20I*, M36I, N37D, R41K, C67S, H69R, V82I, L89M |
| NG_AN.12_06 | G | I13V, K14R, *K20I*, E35Q, M36I, N37S, R41K, K45R, R57K, L63S, C67E, H69K, V82I, L89M |
| NG_AN.12_07 | CRF02_AG | I13A, *K20I*, M36I, R41K, H69K & L89M |
| NG_AN.12_08 | CRF02_AG | *L10V*, I13V, L19P, *K20I*, E35D, M31I, R41K, H69K, K70R, L89M |
| NG_AN.12_09 | G | L10M, I13V, *K20I*, E35Q, M36I, R41K, R57K, I62V, L63P, I64L, C67E, H69K, V82I, L89M |
| NG_AN.12_10 | CRF02_AG | I13V, *K20I*, M36I, N37S, L63S, H69K, K70R, I72M, L89M |
| NG_DE.12_01 | CRF02_AG | I13V, K14R, G16E, *K20I*, E35D, M36I, R41K, H69K, L89M |
| NG_DE.12_02 | CRF02_AG | I13V, K14R, G16E, *K20I*, M36I, R41K, H69K, K70R, I72T, L89M |
| NG_DE.12_03 | G | I13V, *K20I*, E35Q, M36I, R41K, R57K, L63P, C67E, H69K, V82I, L89M, Q92W |
| NG_DE.12_04 | CRF02_AG | *L10I*, I13V, K14R, I15V, L19I, *K20I*, M36I, R41K, I64M, H69K, K70R, L89M |
| NG_DE.12_05 | CRF02_AG | I13V, *K20I*, E35D, M36I, R41K, H69K, K70R, L89M, L97K, F99S |
| NG_DE.12_06 | CRF02_AG | I13V, K14R, *K20I*, M36I, R41K, I64L, E65D, H69K, L89M |
| NG_DE.12_07 | CRF02_AG | I13V, K14R, G17E, *K20I*, M36I, R41K, H69K, L89M |
| NG_DE.12_08 | G | *L10I*, I13V, K14R, *K20I*, E35Q, M36I, R41K, R57K, L63Q, C67E, H69K, V82I, L89M |
| NG_DE.12_09 | U^G | I13V, *K20I*, E35Q, M36I, N37E, L63S, C67E, H69R, V82I, L89M |
| NG_DE.12_10 | CRF02_AG | *L10V*, T12A, I13V, I15V, L19P, *K20I*, M36I, R41K, K43R, L63P, H69K, K70R, L89M |
| NG_IM.12_01 | CRF02_AG | *L10V*, I13V, G16E, *K20I*, E35D, M36I, R41K, I64M, H69K & L89M |
| NG_IM.12_02 | G | I13V, K14R, *K20I*, E35Q, M36I, N37D, R41K, R57K, C67E, H69R, V82I, L89M |
| NG_IM.12_03 | CRF02_AG | I13V, K14R, *K20I*, M36I, R41K, I64L, H69K, L89M |
| NG_IM.12_04 | CRF02_AG | I13V, K14R, *K20I*, E35D, M36I, N37S, R41K, L63S, H69K, L89M |
| NG_IM.12_05 | CRF02_AG | I13V, L19P, *K20I*, *L33F*, M36L, N37H, L38I, R41K, K43R, R57K, L63P, H69K, L89M |
| NG_IM.12_06 | U^G | I13V, G17E, *K20I*, E35Q, M36I, N37E, R57K, L63S, C67E, H69R, V82I, L89M |
| NG_IM.12_07 | CRF02_AG | *L10I*, I13V, K14R, *K20I*, M36I, N37S, R41K, L63S, H69K, I72V, P79S, **V82L**, G86L, R87L, *N88D*, L89M, T91C, Q92H, G94I, T96A, L97V, N98H |
| NG_IM.12_08 | G | I13V, *K20I*, E35Q, M36I, R41K, K45R, R57K, L63P, C67E, H69K, V82I, L89M |

**Key:** PI Major Resistance Mutations are in bold face, PI Minor Resistance Mutations are in italics, and other mutations are in regular face.

by Odaibo *et al.* [17] on pattern of HIV-1 drug resistance among adults on ART in Nigeria. However, some other studies conducted in different parts of the country on both drug naïve and experienced patients reported no major PI resistance mutations [18, 19]. Acquisition of PI resistance is known to be cumulative in nature requiring sequential accumulation of mutations in the setting of on-going exposure to non-suppressive PI-based ART [27, 28]), therefore the appreciable level of IAS PR mutation detected among the patients in our study is a serious cause for concern as it places them at increased risk of accumulating additional PI resistance mutations.

M46I/L is a nonpolymorphic PI-selected mutation that reduces susceptibility to indinavir (IDV), nelfinavir (NFV), fosamprenavir (FPV), lopinavir (LPV) and atazanavir (ATV) when present with other mutations. M46L also reduces susceptibility to tipranavir (TPV). This mutation which occurred at a frequency of 7.14% among all the isolates and frequency of 16.7% among the subtype G isolates accounted for over 66.0% of all the major PI resistance mutations identified in this study. Again, this is also similar to the report of Odaibo *et al.* [17] which reported this mutation at a frequency of 55.6%. Although our study did not determine

**Table 2. Frequency of occurrence of mutations and/or polymorphisms in protease by HIV subtypes.**

| Mutation | No. (%) of mutations | | Mutation | No. (%) of mutations | |
|---|---|---|---|---|---|
| | Subtype G & U$^G$ (n = 12) | CRF02_AG (n = 16) | | Subtype G & U$^G$ (n = 12) | CRF02_AG (n = 16) |
| L10V/I | 1 (8.3) | 6(50.0) | L63T/P/S/Q | 10(83.3) | 5 (31.3) |
| L10M | 1(8.3) | - | I64L/M | 1(8.3) | 4(25.0) |
| T12A | - | 1(6.3) | E65D | - | 1(6.3) |
| I13V/A | 12(100.0) | 16(100.0) | C67E/S | 12(100.0) | - |
| K14R | 6(50.0) | 7(43.8) | H69K/R | 12(100.0) | 16(100.0) |
| I15V | - | 2(12.5) | K70R | 2(16.7) | 6(37.5) |
| G16E | - | 3(18.8) | I72M/T/V | - | 3(18.8) |
| G17E | 1(8.3) | 1(6.3) | P79S | - | 1(6.3) |
| L19P | - | 5(31.3) | V82I | 12(100.0) | - |
| K20I | 12(100.0) | 16(100.0) | V82L | - | 1(6.3) |
| L33F | - | 1(6.3) | G86L | - | 1(6.3) |
| E35Q | 11(91.7) | - | R87L | - | 1(6.3) |
| E35D | - | 6(37.5) | N88D | - | 1(6.3) |
| M36I/L | 12(100.0) | 16(100.0) | L89M | 12(100.0) | 16(100.0) |
| N37D/S/E/H | 5(41.7) | 4 (25.0) | T91C | - | 1(6.3) |
| L38I | - | 1(6.3) | Q92W/H | 1(8.3) | 1 (6.3) |
| R41K | 10(83.3) | 15(93.8) | G94R/I | - | 2(12.5) |
| K43R | 2(16.7) | 2(12.5) | C95W | - | 1(6.3) |
| K45R | 2(16.7) | - | T96A | - | 1(6.3) |
| M46L | 2(16.7) | - | L97K/V | -' | 2(12.5) |
| R57K/G | 10(83.3) | 2(12.5) | N98H | - | 2(12.5) |
| I62V | 1(8.3) | - | F99S | - | 1(6.3) |

**Keys:** Numbers correspond to amino acid positions. The first letter corresponds to the wild-type amino acid; the substituted amino acid is coded by the last letter.

the phenotypic resistance pattern in the infected individuals (a limitation of the study), analysis according to the Stanford algorithm showed that M46L mutation confers potential low-level resistance to ATV/r, FPV/r, IDV/r, LPV/r, TPV/r and intermediate-level resistance to NFV to isolates in this study as shown in Table 3.

Mutations at positions 82 and 88 generally co-exist and result in contraindication to many PIs particularly NFV [29]. In line with this, the only isolate with V82L mutation in this study, NG_IM.12_07, also harbours N88D mutation in addition to L10I and K20I mutations. This V82L mutation is shown to confer low-level resistance to ATV/r, FPV/r and SQV/r as well as potential low-level resistance to IDV/r and LPV/r while it confers intermediate-level resistance to NFV and TPV/r. All the patients had preserved susceptibility to DRV/r since it is the only PI drug analysed that was not selected by any resistance mutation. Similar observations had been reported in a more widespread study in Nigeria which examined the impact of maintaining patients on failing second line ART on the accumulation of PR mutations [30]. Isolates in this study also showed high level of susceptibility (96.4%) to SQV/r. The only isolate with intermediate-level resistance to these drugs had V82L major PI resistance mutation as well as L10I and N88D minor PI resistance mutations. V82L is an uncommon non-polymorphic substrate-cleft mutation known to reduce susceptibility to TPV. This mutation was reported at a frequency of 22.2% in a similar study by Odaibo *et al*. [17].

The different degrees of resistance mutations to the second-line PI drugs in isolates from patients still on first-line ART is worrisome as this is a prelude to treatment failure even before

**Table 3. PI resistance patterns among the HIV isolates from South-eastern Nigeria.**

| Sample | Subtype | Major mutation | Minor mutation | ATV/r | DRV/r | FPV/r | IDV/r | LPV/r | NFV | SQV/r | TPV/r |
|---|---|---|---|---|---|---|---|---|---|---|---|
| NG_AN.12_01 | G | M46L | K20I | P | S | P | P | P | I | S | P |
| NG_AN.12_02 | G | M46L | K20I | P | S | P | P | P | I | S | P |
| NG_AN.12_03 | G | - | K20I | S | S | S | S | S | P | S | S |
| NG_AN.12_04 | CRF02_AG | - | L10V, K20I | S | S | S | S | S | P | S | S |
| NG_AN.12_05 | G | - | K20I | S | S | S | S | S | P | S | S |
| NG_AN.12_06 | G | - | K20I | S | S | S | S | S | P | S | S |
| NG_AN.12_07 | CRF02_AG | - | K20I | S | S | S | S | S | P | S | S |
| NG_AN.12_08 | CRF02_AG | - | L10V, K20I | S | S | S | S | S | P | S | S |
| NG_AN.12_09 | G | - | K20I | S | S | S | S | S | P | S | S |
| NG_AN.12_10 | CRF02_AG | - | K20I | S | S | S | S | S | P | S | S |
| NG_DE.12_01 | CRF02_AG | - | K20I | S | S | S | S | S | P | S | S |
| NG_DE.12_02 | CRF02_AG | - | K20I | S | S | S | S | S | P | S | S |
| NG_DE.12_03 | G | - | K20I | S | S | S | S | S | P | S | S |
| NG_DE.12_04 | CRF02_AG | - | L10I, K20I | S | S | S | S | S | P | S | S |
| NG_DE.12_05 | CRF02_AG | - | K20I | S | S | S | S | S | P | S | S |
| NG_DE.12_06 | CRF02_AG | - | K20I | S | S | S | S | S | P | S | S |
| NG_DE.12_07 | CRF02_AG | - | K20I | S | S | S | S | S | P | S | S |
| NG_DE.12_08 | G | - | L10I, K20I | S | S | S | S | S | P | S | S |
| NG_DE.12_09 | U$^G$ | - | K20I | S | S | S | S | S | P | S | S |
| NG_DE.12_10 | CRF02_AG | - | L10V, K20I | S | S | S | S | S | P | S | S |
| NG_IM.12_01 | CRF02_AG | - | L10V, K20I | S | S | S | S | S | P | S | S |
| NG_IM.12_02 | G | - | K20I | S | S | S | S | S | P | S | S |
| NG_IM.12_03 | CRF02_AG | - | K20I | S | S | S | S | S | P | S | S |
| NG_IM.12_04 | CRF02_AG | - | K20I | S | S | S | S | S | P | S | S |
| NG_IM.12_05 | CRF02_AG | - | K20I, L33F | S | S | P | S | S | L | S | P |
| NG_IM.12_06 | U$^G$ | - | K20I | S | S | S | S | S | P | S | S |
| NG_IM.12_07 | CRF02_AG | V82L | L10I, K20I N88D | L | S | L | P | P | I | L | I |
| NG_IM.12_08 | G | - | K20I | S | S | S | S | S | P | S | S |

ATV = Atazanavir; DRV = Darunavir; FPV = Fosamprenavir; IDV = Indinavir; LPV = Lopinavir; NFV = Nelfinavir; SQV = Saquinavir; TPV = Tipranavir; r = ritonavir; S, P, L and I indicate Susceptible, Potential low-level, Low-level and intermediate-level resistant to drugs respectively

switching to second-line ART. This further limits the choice of treatment regimen available for patients failing first-line therapy when the need arises.

Our study also revealed very high frequencies of minor mutations in the *protease* gene of the isolates with the predominant mutations found at positions L10I/V and K20I. Similar finding was reported in Jos, North-Central Nigeria among isolates from ART-naïve patients [16]. Although the presence of these minor mutations do not lead to high level resistance when occurring alone, they have to be taken into account by physicians before making treatment decisions as they may play a role in improving viral fitness or increasing the drug resistance level in the presence of major PI mutations [31–34].

*Pol* gene polymorphisms usually occur in non-B HIV-1 strains as genetic fingerprints that lowers their susceptibility to ARV compounds [35–37]. Other mutations/polymorphisms that occurred at very high frequencies among patients in this study include; I13V/A, K20I, E35Q, M36I/L, R41K, R57K/G, L63T/P/S/Q, C67E/S, H69K/R, V82I and L89M. Similar mutations/ polymorphic substitutions in the protease region had been reported earlier for some Nigerian isolates [13]. I13V/A, K20I, M36I/L, R41K, H69K/R and L89M are the consensus mutations

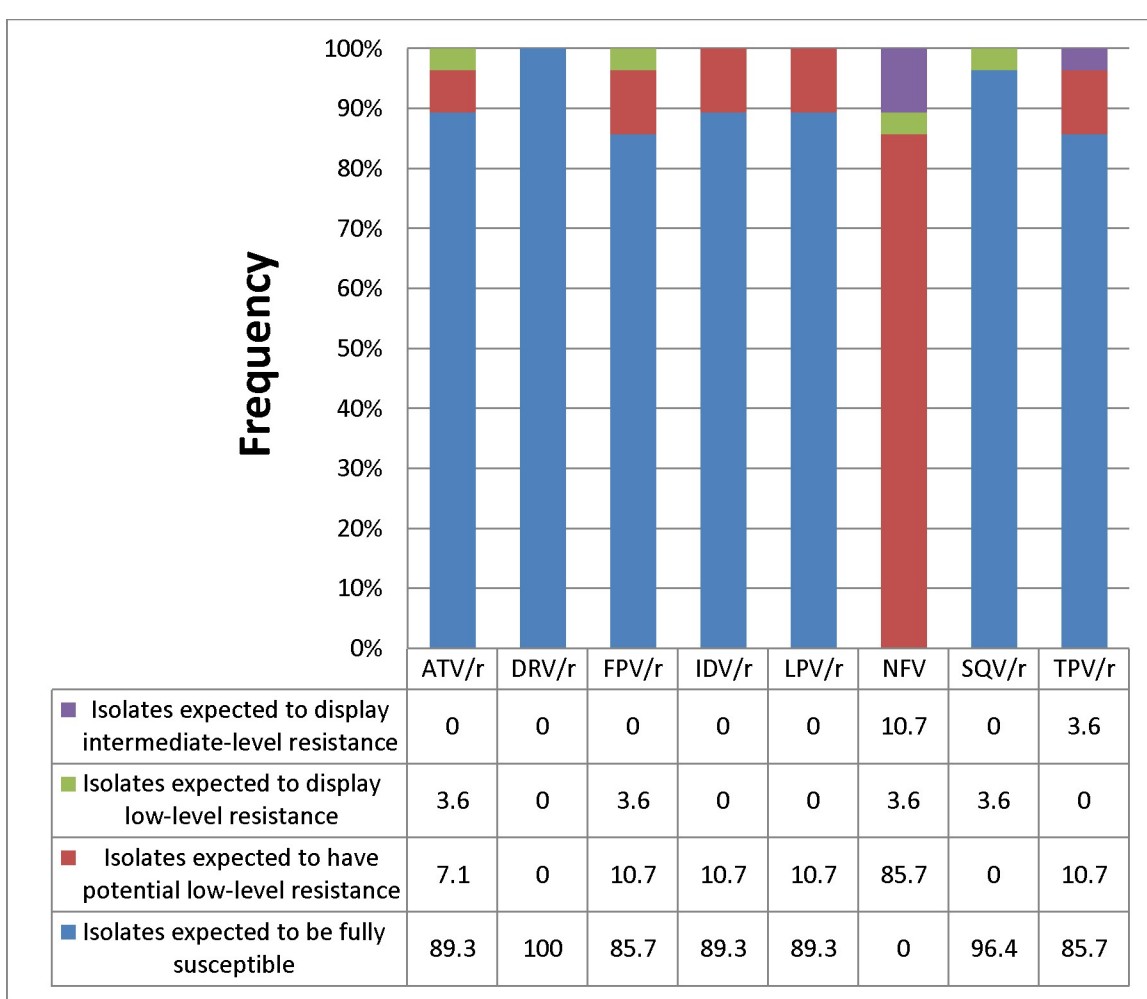

**Fig 3. Predicted susceptibility of the isolates to protease inhibitors (PIs).** ATV = Atazanavir, DRV = Darunavir, FPV = Fosamprenavir, IDV = Indinavir, LPV = Lopinavir, NFV = Nelfinavir, SQV = Saquinavir, TPV = Tipranavir, r = Ritonavir.

identified for subtypes G, U$^G$ and CRF02_AG while E35Q, R57K/G, C67E/S and V82I are the consensus mutations for G and U$^G$ in this study. A study on PI-naive Nigerian HIV-patients have earlier identified I13V, M36I and H69K as wild-type consensus mutations for HIV-1 subtypes G′, G, CRF02_AG, CRF06_cpx, and A. The same study also identified K20I as consensus for G′, G, CRF02_AG, and CRF06_cpx while V82I was identified as the consensus for G′ and G [14]. Although our study utilized samples from first-line drug experienced individuals, the findings are consistent with these earlier reports from drug naïve individuals. In addition, the mutations K14R, N37D/S/E/H and L63T/P/S/Q occurred in ≥ 25% of subtype G, U$^G$ and CRF02_AG patients, at a proportion that is significantly greater than in subtype B. Similarly, mutations L10V/I, L19P, E35D, I64L/M and K70R occurred in ≥ 25% of CRF02_AG patients at a proportion that is significantly greater than in subtype B in this study. There is already a concern that treatment of non-subtype B infected persons with PI could be less effective as a result of higher frequency of polymorphism in the protease gene of non-B isolates including positions 20, 36, 63 and 82 [10]. The limitation of this study however is the small sample size (28 samples) analysed. A larger sample size would have showed a clearer picture of the situation.

In conclusion, we have shown that major and minor PI drug resistance mutations occur in significant proportions of non-B HIV-1 strains circulating among first-line drug experienced individuals in south-eastern Nigeria. This study also demonstrated differences in the distribution pattern of these mutations between subtypes G and CRF02_AG isolates. The high level of these drug resistance mutations will not augur well for PI treatment interventions. This result therefore underscores the need for periodic genotypic DR testing of patients on ART and prior to second-line ART switch for early detection of DR mutations and selection of appropriate treatment regimens. The only challenge however is the high cost of carrying out these tests which may limit its implementation in resource limited settings.

## Acknowledgments

We appreciate the management of General Hospital Awo-Omamma, Imo state; State Hospital Asaba, Delta state and St Joseph's Catholic Hospital Adazi, Anambra state for facilitating access to the patients in this study.

## Author Contributions

**Conceptualization:** Augustine O. Udeze, David O. Olaleye, Georgina N. Odaibo.

**Data curation:** Augustine O. Udeze.

**Formal analysis:** Augustine O. Udeze.

**Funding acquisition:** Augustine O. Udeze.

**Methodology:** Augustine O. Udeze, David O. Olaleye, Georgina N. Odaibo.

**Supervision:** David O. Olaleye, Georgina N. Odaibo.

**Writing – original draft:** Augustine O. Udeze.

**Writing – review & editing:** Augustine O. Udeze, David O. Olaleye, Georgina N. Odaibo.

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
