## [Decision Letter · Decision Letter 0]

4 Sep 2019

PONE-D-19-17545

Polymorphisms and drug resistance analysis of HIV-1 isolates from patients on first line antiretroviral therapy (ART) in South-eastern Nigeria

PLOS ONE

Dear Dr. Odaibo,

Thank you for submitting your manuscript to PLOS ONE. After careful consideration, we feel that it has merit but does not fully meet PLOS ONE’s publication criteria as it currently stands. Therefore, we invite you to submit a revised version of the manuscript that addresses the points raised during the review process.

We would appreciate receiving your revised manuscript by Oct 19 2019 11:59PM. To enhance the reproducibility of your results, we recommend that if applicable you deposit your laboratory protocols in protocols.io, where a protocol can be assigned its own identifier (DOI) such that it can be cited independently in the future. For instructions see: http://journals.plos.org/plosone/s/submission-guidelines#loc-laboratory-protocols

We look forward to receiving your revised manuscript.

Kind regards,

Jason Blackard, PhD

Academic Editor

PLOS ONE

Journal Requirements:

2. Our internal editors have looked over your manuscript and determined that it is within the scope of our Antimicrobial Resistance call for papers. This collection of papers is headed by a team of Guest Editors for PLOS ONE: Kathryn Holt (Monash University and London School of Hygiene and Tropical Medicine), Alison H. Holmes (Imperial College London), Alessandro Cassini (WHO Infection Prevention and Control Global Unit), Jaap A. Wagenaar (Utrecht University). The Collection will encompass a diverse range of research articles; additional information can be found on our announcement page: https://collections.plos.org/s/antimicrobial-resistance. If you would like your manuscript to be considered for this collection, please let us know in your cover letter and we will ensure that your paper is treated as if you were responding to this call. If you would prefer to remove your manuscript from collection consideration, please specify this in the cover letter.

3. Please specify in your ethics statement: 1) whether the ethics committee approved the verbal/oral consent procedure, 2) why written consent could not be obtained, and 3) how verbal/oral consent was recorded.” Do not ping with follow up unless there are questions, in which case, ping me.

Additional Editor Comments (if provided):

This a cross-sectional of HIV drug resistance in persons on first-line ART in Nigeria.

While 3 study sites were include, the overall population size is modest.  It is unclear why only a very short study period (February to May 2012) was included.

Have the authors explored the two “unclassifiable” sequences in more detail?  What are the closest hits in a Blast search?

For Table 2, were there any statistically significant differences in mutations when the data are stratified by subtype?

If the point of Figure 2 is to show polymorphism at particular amino acid sites, creating a WebLogo for this region would better illustrate this.

It is unclear from the introduction what specific antiretroviral therapies were available in this region of Nigeria during the 2012 study period.

Limitations of the current study should be described in more detail in the Discussion.

Reviewers' comments:

Reviewer's Responses to Questions

**Comments to the Author**

1. Is the manuscript technically sound, and do the data support the conclusions?

Reviewer #1: Yes

Reviewer #2: Partly

2. Has the statistical analysis been performed appropriately and rigorously? 

Reviewer #1: N/A

Reviewer #2: Yes

3. Have the authors made all data underlying the findings in their manuscript fully available?

Reviewer #1: Yes

Reviewer #2: Yes

4. Is the manuscript presented in an intelligible fashion and written in standard English?

Reviewer #1: Yes

Reviewer #2: Yes

5. Review Comments to the Author

Reviewer #1: ART use has greatly reduced morbidity and mortality among people living with HIV, and subsequently improved life expectancy and quality of life.

However, genetic diversity and specific genetic resistance pathways could impair these individual and community ART benefits in resource-limited settings.

This study aimed to describe resistance patterns to PIs of HIV isolates from first-line drug experienced individual in Imo state (Nigeria). This study was conducted on 28 HIV-1 infected patient enrolled between February and may 2012 in a general hospital.

Mains results of this study highlighted a significant polymorphism in HIV protease resistance gene. According to the Stanford drug-resistance algorithm, in this study, all the PIs excepted darunavir were affected by this important polymorphism and two non-polymorphic mutation M46L and V82L. Two HIV-1 non B subtypes G and CRF02_AG were described with differences in the pattern of these polymorphic mutations.

Overall, this study reported the genetic diversity of HIV-1 in Nigeria and assessed the prevalence of the PIs drug resistance mutation in first ART first line experience individuals. PIs regimen outcomes in resource-limited settings can be impacted by the high prevalence of the genetic diversity and point out the necessity of genotypic resistance assays before switching to second-line ART.

Strengths and weaknesses

The research was conducted in ethical and sound research background. This paper is well organized and followed the manuscript guidelines of the journal to a large extent. This study confirms that in resource-limited west African countries, HIV-1 non-B subtypes are majority and have significant polymorphism on protease resistance codons, as described by several authors.

The inclusion criteria should be more detailed, especially on the therapeutic aspects (patients being treated).

The number of samples included should be explained.

Details on the choice of the DNA matrix would be important additional information for the article understanding

Please mention primers used reference.

Reviewer #2: The study reported on PI drug resistance mutations that were present at first line failure in a cohort of patients from 3 states in south-eastern Nigeria. The study highlights the fact that major and minor PI mutations were found before the initiation of a PI-inclusive 2nd-line, with DRV the only fully active PI available.

While the data does contribute to the knowledge of circulating subtypes in the region, a short-coming of the study is that the samples size is very small, especially since it is meant to be representative of the south-eastern region. The study would therefore benefit from an increased sample size. In addition, the naturally occurring polymorphisms in protease that are reported are already known for these subtypes.

Specific comments:

Line 117-118: Informed consent was already mentioned in line 116. Is this meant to be different?

Line 148-150: it is not clear why these older versions of the drug resistance lists are cited here.

Line 179: should be "L89 (100%)"

Line 194: The V82I mutation is already known to occur in subtype G isolates.

Line 324: "high level" is misleading as this occurs in only 2 isolates.

Generally there are grammatical throughout the paper an would benefit from a language editor. Some are listed below.

Line 55: should read "....multiple circulating subtypes."

Line 67: "...ARV therapy..."

Line 136: should read "...downloaded from the Los Alamos...."

Line 146: should read "...protease gene..."

Line 325: cause not "course"

Line 327: should read "M46I/L is a nonpolymorphic PI-selected mutation that reduces...."

Line 344: should read "....PI drug analysed …resistance mutation."

Line 345 "observations"

Line 354: "switching to..."

6. PLOS authors have the option to publish the peer review history of their article (what does this mean?). If published, this will include your full peer review and any attached files.

Reviewer #1: No

Reviewer #2: No

---

## [Author Response · Author response to Decision Letter 0]

27 Dec 2019

Reviewer #1

Comment: The inclusion criteria should be more detailed, especially on the therapeutic aspects (patients being treated).

Response: Done (lines 112-113)

Comment: Please mention primers used reference.

Response: Done (line 127)

---

## [Decision Letter · Decision Letter 1]

16 Mar 2020

Polymorphisms and drug resistance analysis of HIV-1 isolates from patients on first line antiretroviral therapy (ART) in South-eastern Nigeria

PONE-D-19-17545R1

Dear Dr. Odaibo,

We are pleased to inform you that your manuscript has been judged scientifically suitable for publication and will be formally accepted for publication once it complies with all outstanding technical requirements.

With kind regards,

Jason Blackard, PhD

Academic Editor

PLOS ONE

Additional Editor Comments (optional):

None

Reviewers' comments:

None

Reviewer's Responses to Questions

**Comments to the Author**

1. If the authors have adequately addressed your comments raised in a previous round of review and you feel that this manuscript is now acceptable for publication, you may indicate that here to bypass the “Comments to the Author” section, enter your conflict of interest statement in the “Confidential to Editor” section, and submit your "Accept" recommendation.

Reviewer #1: All comments have been addressed

2. Is the manuscript technically sound, and do the data support the conclusions?

Reviewer #1: Yes

3. Has the statistical analysis been performed appropriately and rigorously? 

Reviewer #1: Yes

4. Have the authors made all data underlying the findings in their manuscript fully available?

Reviewer #1: Yes

5. Is the manuscript presented in an intelligible fashion and written in standard English?

Reviewer #1: Yes

6. Review Comments to the Author

Reviewer #1: All comments have been addressed specially on sample size, the method and the therapeutic regimen. The paper is good for publication.

7. PLOS authors have the option to publish the peer review history of their article (what does this mean?). If published, this will include your full peer review and any attached files.

Reviewer #1: No

---

## [Editor Report · Acceptance letter]

23 Mar 2020

PONE-D-19-17545R1 

Polymorphisms and drug resistance analysis of HIV-1 isolates from patients on first line antiretroviral therapy (ART) in South-eastern Nigeria 

Dear Dr. Odaibo:

I am pleased to inform you that your manuscript has been deemed suitable for publication in PLOS ONE. Congratulations! Your manuscript is now with our production department. 

With kind regards,

on behalf of

Dr. Jason Blackard 

Academic Editor

PLOS ONE